# Research on the Coupling Coordination of Green Finance, Digital Economy, and Ecological Environment in China

**Lifang Zhang and Yuexu Zhao \***

College of Economics, Hangzhou Dianzi University, Hangzhou 310018, China; zhanglifang980320@163.com
\* Correspondence: yxzhao@hdu.edu.cn

**Abstract:** This study analyzes the coupling coordination of green finance, digital economy, and ecological environment, and constructs an evaluation index system of coupling coordination degree. Based on the panel data of 30 provinces in China from 2011 to 2020, this study applies the coupling coordination model, spatial autocorrelation model, and gray correlation model to analyze the spatio-temporal evolution characteristics of coupling coordination degree and driving factors. The results indicate that the overall level of green finance, digital economy, and ecological environment maintains steady development, among them, the digital economy is developing the fastest. The coupling coordination degree among the three subsystems exhibits an ascending trend and transitions from dissonance to coordination and displays significant global and local spatial autocorrelation characteristics. Regional disparities exist between the driving factors that influence the coupling coordination degree. Therefore, the existing green financial system should be optimized, coordination of green finance and digital economy synergies should be improved, and each region should devise a development strategy tailored to its regional characteristics.

**Keywords:** green finance; digital economy; ecological environment; coupling coordination

## 1. Introduction

With the rapid development of the economy, problems such as natural resource depletion and environmental contamination are intensifying, and the conflict between rapid economic development and environmental protection is deepening. Green finance serves as an essential link between economic development and environmental governance. The purpose of green finance is to achieve sustainable development between the economy and the environment, and the redirected green finance funds can stimulate businesses to innovate and create [1–3], reduce environmental contamination [4], and achieve coordinated synergistic growth of the environment and finance [5].

In recent years, the digital economy has transitioned from the traditional model of economic development to the modern model of high-quality economic development. High technology, high integration, and high potential are hallmarks of the digital economy, which plays a central role in enhancing the development of the environment. In addition to promoting the scope and depth of the economy, the digital economy can also facilitate the coordinated development of economic and environmental development [6]. As a modern new economic technology and convergence economy, the digital economy contributes significantly to the development of a low-carbon and green environment [7,8].

Green finance and the digital economy represent a crucial direction for sustainable development. Green finance is the new propelling force of economic development, and the ecological environment is the resource guarantee for sustainable economic development. The three systems of green finance, digital economy, and ecological environment (henceforth referred to as the GDE system) influence and constrain each other, and sustainable development must account for their synergistic development.

The existing literature focuses primarily on the economic and ecological environment, as well as the effects of green financial development on the economy and the environment.

Few academics have examined the relationship between green finance and the digital economy, and even fewer have investigated the coupling coordination of green finance, digital economy, and ecological environment. In response to this research need, this study endeavors to develop an evaluation index system for measuring the spatio-temporal evolution characteristics of the coupling coordination degree of GDE systems and driving factors.

The remainder of the paper is organized as follows: Section 2 gives a literature review of this article. Section 3 presents the methodology. Section 4 analyses the spatio-temporal evolution characteristics of coupling coordination of the GDE system. Section 5 studies the driving factors of the coupling coordination of the GDE system. Finally, Section 6 concludes the research and gives policy suggestions.

## 2. Literature Review

The studies related to green finance, digital economy, and ecological environment can be summarized into the following two major aspects. Firstly, in economic and ecological environment-related research, Yousaf et al. [9] explored the direct impact of the digital orientation, Internet of Things (IoT), and digital platforms on sustainable digital innovation in the context of the digital economy and frugal environment. Shahbaz et al. [10] investigated the effect of the digital economy on the structure of renewable energy consumption. Aminullah et al. [11] examined the interactive components of the digital micro and small medium enterprise (MSMEs) ecosystem for an inclusive digital economy. Yu et al. [12] adopted a multistage DID model to evaluate the impact of the digital economy on high-quality economic development and found that the digital economy has a significant role in promoting high-quality economic development. Lee et al. [13] addressed the issue of the relationship between economic growth and environmental quality in China and confirms the inverse U-shaped relationship as well as the N-shaped relationship between income and pollution. Huang et al. [14] analyzed the impact of the digital economy on urban environmental quality. Li et al. [15] studied the degree of coupling coordination between the digital economic system and the environmental system. Fu et al. [16] provided a basis for understanding the spatial and temporal evolution characteristics of the coordinated development of China's digital economy and ecological environment. Li et al. [17–19] explored the influence of filtrate reducer and reservoir characteristics on the filtration reduction of drilling fluid during the drilling process.

Second, the impact of green finance development on the economy and the environment is studied. Sinha et al. [20] empirically analyzed the impact of green bond financing on environmental and social sustainability. Mngumi et al. [21] investigated the links between green finance, renewable energy, and $CO_2$ emissions. Meo et al. [22] examined the relationship between green finance and carbon dioxide ($CO_2$) emissions in the top ten economies that support green finance. Irfan et al. [23] empirically tested the influence mechanism and policy intervention effects of inclusive green finance on green innovation. Yang et al. [24] pointed out that green finance has significantly improved China's ecological and livable environment by improving the level of technological innovation. Li and Gan [25] constructed a spatial Dubin model and the empirical results show that the development of green finance promotes the improvement of the ecological environment in this region and the influence of green finance on the ecological environment has a significant positive spatial spillover effect. Zhu [26] addressed the coupling coordination degree between green finance and marine eco-environment, and the results showed that the coupling coordination between green finance and the ecological environment system are both slowly increasing. Tang et al. [27] explored the impact of green finance on the quality of the ecological environment in the Yangtze River Economic Belt. Fu and Irfan [28] investigated the influence of green finance and financial development on environmental sustainability and growth in ASEAN economies from 2012 to 2019. Zhan et al. [29] studied the impact of green finance and financial innovation on the environmental status in China from 1996 to 2020.

In conclusion, the existing literature focuses primarily on the study of the economic and ecological environment, as well as the effect of green financial development on the economy and environment. Few scholars examine the relationship between green finance and the digital economy, and even fewer examine the coupling coordination between the GDE system. Insufficient research has been conducted on the law of coupling coordination development and further analysis of the characteristics of their dynamic evolution from the perspective of spatio-temporal coupling. The scope of the research is limited to a single region or province, with no inter-provincial comparisons on a mesoscopic scale.

The following three points are expanded on in this article when compared to the prior literature: (1) The existing literature primarily focuses on the level of coupling coordination development of single systems or double systems, but this paper makes use of the findings of previous studies, integrates green finance, the digital economy, and the ecological environment, examines the coupling coordination of the GDE system, and attempts to create a comprehensive level evaluation index system for the three. (2) To completely express the spatial and temporal evolution characteristics of the GDE system, the modified coupling coordination model is used to measure and assess the coupled coordination development of the GDE system. (3) This paper explores the three driving factors of the coupling coordination of the three subsystems after empirically analyzing the rising trend of coupling coordination fluctuation and uneven spatial distribution of the GDE system. The purpose of this paper is to present a scientific foundation to supply relevant departments to formulate policies regarding green finance, the digital economy, and the ecological environment, and to promote the coupling coordination of the GDE system.

## 3. Methodology

### 3.1. Mechanistic Analysis

Multiple systems can interact and consequently influence one another through coupling. The degree of coupling coordination reflects both the developmental stage of each system and the nature of their mutual influence and coordinated growth. The coupling coordination development between systems and the changing trend has an immense effect on ecological environmental protection and high-quality development in China, and the coordination development of the GDE system is a crucial foundation for promoting the green transformation of the economy.

Green finance can facilitate the growth of the digital economy. Green development is the basis for dealing with environmental problems and is a necessary demand for building a green economic system in China. In the current context of green development, attaining carbon peaks in China is predicated on the transformation and upgrading of traditional industries, creating landing scenarios and development directions for the transformation of traditional industries as well as industrial digitalization [30]. Green finance can be combined with high energy consumption digital economy industries to vigorously promote sustainable technological innovation and reduction of energy consumption in related industries, and thus support the development of the digital economy.

The growth of the digital economy facilitates the expansion of green finance. The digital economy promotes the development of green finance through its "precision". The digital economy mainly applies artificial intelligence, cloud computing, and other advanced technologies to establish a network-type open ecological system, which opens up the basic links between the whole industry for green finance, forms an information resource sharing system, ensures the security and validity of data, effectively improves the efficiency of investment and financing matching, reduces the risks and losses caused by information mismatch, and thus continuously expands the scale of investment and financing, and helps green finance effectively integrate the industrial chain and value chain.

The ecological environment is a prerequisite for the robust growth of green finance and the digital economy. A good ecological environment and sufficient natural resources can continuously absorb financial resources to join the construction process, thereby optimizing China's financial structure, further enhancing financial services, and effectively promoting

the development of green finance. The development of the coupling coordination of green finance, digital economy, and eco-environmental has a significant effect on China's high-quality development, so it is important to measure and analyze the coupled and coordinated degree of the GDE system.

### 3.2. Construction of Evaluation Index System

(1) Green finance. Before 2010, China's information concerning green finance was in a fragmented state, and data collection was difficult. After 2010, the disclosure of pertinent information began to be progressively standardized, and the quality of data indicators gradually remained the same. The basic principle is to highlight the core of green finance, based on relevant studies [31–33], credit, investment, securities, insurance, carbon 6 indicators in 5 dimensions of finance (Table 1), and use the entropy value method to assess the degree of green finance development in China, to make a comprehensive and scientific evaluation.

**Table 1.** Evaluation index system of green finance, digital economy, and ecological environment.

| System | Levels | Indicators | Characteristics of Indicators |
|---|---|---|---|
| Green Finance | Green Credit | High energy-consuming industrial industry interest expenses/Total industrial interest expenses | − |
| | Green Investment | Environmental pollution control investment/GDP | + |
| | | Fiscal expenditure on energy conservation and environmental protection/Total fiscal expenditure | + |
| | Green Securities | The total output value of environmental protection enterprises/Total A-share market capitalization | + |
| | Green Insurance | Agricultural premium income/Agricultural output | + |
| | Carbon Finance | Carbon finance transaction amount/Total national carbon finance transaction | + |
| Digital Economy | Digital Industrialization | Percentage of employment in information transmission, computer services, and software | + |
| | | Log software business revenue | + |
| | | Information transmission, computer services, and the software industry accounted for the proportion of fixed assets in society as a whole | + |
| | Industry Digitization | Industrial value added | + |
| | | Tertiary industry value added | + |
| | | Value added of agriculture, forestry, animal husbandry, and fishery | + |
| | | Taobao village number percentage | + |
| | | Digital inclusive finance index | + |
| | Digital Infrastructure | Number of domains | + |
| | | Number of Internet users | + |
| | | Number of websites | + |
| | | Internet penetration rate | + |
| | | Cell phone penetration rate | + |
| Ecology | Living Environment | Green space per capita | + |
| | | Forest cover | + |
| | | Greening coverage of built-up areas | + |
| | Energy Consumption | Electricity consumption per unit of GDP | − |
| | | Amount of energy consumption per unit of GDP | − |
| | Pollution Intensity | Industrial wastewater discharge | − |
| | | Industrial Sulfur dioxide emissions | − |
| | | Industrial dust emissions | − |
| | Treatment Intensity | The comprehensive utilization rate of industrial solid waste | + |
| | | Domestic wastewater treatment rate | + |
| | | Investment in environmental management as a percentage of GDP | + |
| | | $SO_2$ removal rate | + |

(2) Digital economy. In the existing literature, two primary approaches to measuring the digital economy are identified. (i) Using a single indicator to estimate the extent of digital economy development within a specified range. (ii) Using a multiple indicator system to obtain the relative situation of digital economy development in each region through comparison. However, the focus of these digital economy indicators measures are

somewhat different, and there are problems such as unstable data sources, restricted data selection, poor operability, insufficient comparability, narrow application breadth and short period, and poor sample extensibility. This paper develops a digital economy index based on the definition of the digital economy, focusing on the mean quantity and referring to existing studies [34–36]. Under the three dimensions of digital industrialization, industrial digitization, and digital infrastructure, the index measures a total of thirteen indicators.

(3) Ecological environment. The construction of ecological civilization is a basic condition for high-quality development and its important outcome. Adhering to green development and reducing pollution emissions has been one of the focuses of global economic policies. After referring to studies in the related literature [3,37,38], considering the availability of data, twelve indicators in four dimensions of the living environment, energy consumption, pollution intensity, and governance intensity are selected to measure the level of ecological environment comprehensively, which can effectively reflect the main inputs and achievements of the region in managing the environment.

(4) Data source. To conduct the empirical analysis, this paper focuses on the panel data of 30 Chinese provinces (excluding Tibet, Hong Kong, Macau, and Taiwan) from 2011 to 2020. The majority of the data used comes from the EPS database, the CSMAR database, the Wind database, the China Statistical Yearbook, the China Financial Statistical Yearbook, the China Environmental Statistical Yearbook, and the China Insurance Statistical Yearbook, among others. For absent data, linear interpolation using data from the closest year was conducted.

*3.3. Research Methodology*

3.3.1. Entropy Method

This paper employs the entropy method to calculate the relative importance of each indicator, which is more objective compared with the expert scoring method and the hierarchical analysis method. Since the different index levels can lead to large errors in the calculation of the indexes, this paper first uses the threshold method for dimensionless data processing, the following describes how positive and negative indexes are evaluated:

$$\widetilde{x}_{it,j} = \frac{x_{it,j} - \min_{j}\{x_{it,j}\}}{\max_{j}\{x_{it,j}\} - \min_{j}\{x_{it,j}\}} \tag{1}$$

$$\widetilde{x}_{it,j} = \frac{\max_{j}\{x_{it,j}\} - x_{it,j}}{\max_{j}\{x_{it,j}\} - \min_{j}\{x_{it,j}\}} \tag{2}$$

where $\widetilde{x}_{it,j}$ denotes the dimensionless variable for the $j$-th indicator for the $i$-th province in year $t$, the $\max_{j}\{x_{it,j}\}$ and $\min_{j}\{x_{it,j}\}$ denote the maximum and minimum values in each indicator respectively.

Calculate the weight of the $j$-th indicator for the $i$-th province in year $t$:

$$p_{it,j} = \frac{\widetilde{x}_{it,j}}{\sum_{i=1}^{N} \sum_{t=1}^{T} \widetilde{x}_{it,j}} \tag{3}$$

where $N = 30$ is the number of cross sections, and $T = 10$ is the number of years.

Calculate the information entropy and redundancy of the $j$-th indicator:

$$e_j = \frac{1}{\ln(NT)} \sum_{i=1}^{N} \sum_{t=1}^{T} p_{it,j} \ln(p_{it,j}) \tag{4}$$

$$d_j = 1 - e_j \tag{5}$$

where $e_j \in [0, 1]$.

Calculate the weight of the $j$-th indicator based on the redundancy of the information entropy:

$$W_j = \frac{d_j}{\sum_{j=1}^{m} d_j} \tag{6}$$

Finally, the composite index of each system can be calculated using the following formula:

$$U_i = \sum_{j=1}^{n} W_j Y_{ij} \tag{7}$$

where $U_i$ is the integrated level indicator of each system and $n$ is the number of indicators within each system.

### 3.3.2. Modified Coupling Coordination Model

The coupling degree model commonly used by scholars [39–41] is calculated as follows:

$$C = \left[ \frac{\prod_{i=1}^{n} U_i}{\left( \frac{1}{n} \sum_{i=1}^{n} U_i \right)^n} \right]^{\frac{1}{n}} \tag{8}$$

where $n$ represents the number of subsystems, in this paper $n = 3$. $U_i$ represents the value of each subsystem, and the corresponding distribution interval is $[0, 1]$. Therefore, the coupling degree $C$ value interval is $[0, 1]$. The larger the value, the smaller the degree of dispersion between the subsystems, and the higher the coupling degree; the smaller the value, the lower the subsystem coupling degree. Furthermore, the value can be subdivided into four intervals: $(0, 0.3]$ is the low level of coupling, $(0.3, 0.5]$ is the antagonistic state, $(0.5, 0.8]$ is the grinding stage, and $(0.8, 1]$ is the high-level coupling.

However, the assumptions underlying the above coupling coordination model for the analysis of the coupling degree $C$ value are $C \in [0, 1]$ and the $C$ value is uniform over the interval $[0, 1]$ and is uniformly distributed over the interval. However, the problem here is that the validity of the $C$ value and the coordinated development degree model will be simplified [42], so this paper chooses to use a modified coupling degree model with the following formula.

$$C = \sqrt{\left[ 1 - \frac{\sum_{i>j,j=1}^{n} \sqrt{(U_i - U_j)^2}}{\sum_{m=1}^{n-1} m} \right] \times \left( \prod_{i=1}^{n} \frac{U_i}{maxU_i} \right)^{\frac{1}{n-1}}} \tag{9}$$

$$T = \sum_{i=1}^{n} \alpha_i \times U_i, \quad \sum_{i=1}^{n} \alpha_i = 1 \tag{10}$$

$$D = \sqrt{C \times T} \tag{11}$$

Among them: $U_i \in [0,1]$, $C \in [0,1]$. In this paper, we study the relationship between the coupling coordination degree of the three, so we take $n = 3$, $\alpha_1 = \alpha_2 = \alpha_3 = 1/3$ (assigning equal weights), $D$ is the coupling coordination degree, $i, j = 1, 2, 3$. The coupling coordination degree is defined as ten levels (see Table 2) by drawing on the study of Cheng et al. [43].

**Table 2.** Classification of coupling coordination level.

| D | Coordination Level | D | Coordination Level |
|---|---|---|---|
| [0, 0.1) | Extreme dissonance | [0.5, 0.6) | Barely coordination |
| [0.1, 0.2) | Severe dissonance | [0.6, 0.7) | Primary coordination |
| [0.2, 0.3) | Moderate dissonance | [0.7, 0.8) | Intermediate coordination |
| [0.3, 0.4) | Mild dissonance | [0.8, 0.9) | Good coordination |
| [0.4, 0.5) | On the verge of dissonance | [0.9, 1] | Quality coordination |

### 3.3.3. Spatial Autocorrelation

Spatial autocorrelation can effectively measure the degree of clustering and dispersion of spatial unit attribute values and can be divided into global and local spatial autocorrelation. Among them, global spatial autocorrelation is a measure of whether the data has clustering or dispersion in the whole spatial distribution, which is calculated as follows:

$$I = \frac{\sum_{i=1}^{n} \sum_{j=1}^{n} \omega_{ij} (\chi_i - \overline{\chi})(\chi_j - \overline{\chi})}{S^2 \sum_{i=1}^{n} \sum_{j=1}^{n} \omega_{ij}} \tag{12}$$

where $i \neq j$, $n$ represents the number of provinces, and $\chi_i$ represents the number of $i$ provincial observations. $S^2$ represents the sample variance, $\overline{\chi}$ denotes the sample mean, and $\omega_{ij}$ is the spatial weight matrix, with 1 for spatial contiguity and 0 for non-contiguity. The range of values for the Moran index is $[-1, 1]$. If the value is greater than zero, it means there is a positive spatial correlation, and vice versa, it means there is a negative spatial correlation, and a value of zero means there is no spatial correlation.

The local spatial autocorrelation is a measure of whether each spatial element is correlated in local space, which corresponds to the local Moran index calculated as:

$$I_i = \frac{\chi_i - \overline{\chi}}{S^2} \sum_{j=1}^{n} \left[ \omega_{ij} (\chi_i - \overline{\chi}) \right] \tag{13}$$

## 4. Analysis of the Spatio-Temporal Evolution Characteristics of Coupling Coordination of the GDE System

### 4.1. Analysis of the Development Characteristics of the GDE System

Figure 1 demonstrates the outcomes of using the entropy method to calculate the level of green finance, digital economy, and ecological environment. All three comprehensive indices exhibit a consistent upward trend from 2011 to 2020. The green amount index increased from 0.241 in 2011 to 0.341 in 2020 with an average annual growth rate of 3.9%, with the highest growth rate being 4.56% in 2016, which is primarily attributable to China's significant advances in green financial products, tools, and technologies in 2016. China's green bond market was launched in 2016 and rapidly expanded to become the world's largest green bond market, issuing green asset-backed securities and green asset-backed securities. With an average annual growth rate of 6.89%, the digital economy index increased from 0.174 to 0.312, thereby creating crucial conditions for the development of green finance and the ecological environment. The ecological environment index increased from 0.366 to 0.533 at a rate of 5.88% per year on average. It has supported the growth of the regional digital economy and the sustainable development and enhancement of green industries, which is conducive to the expansion of green finance. The digital economy has the highest growth rate, while ecological finance has the lowest.

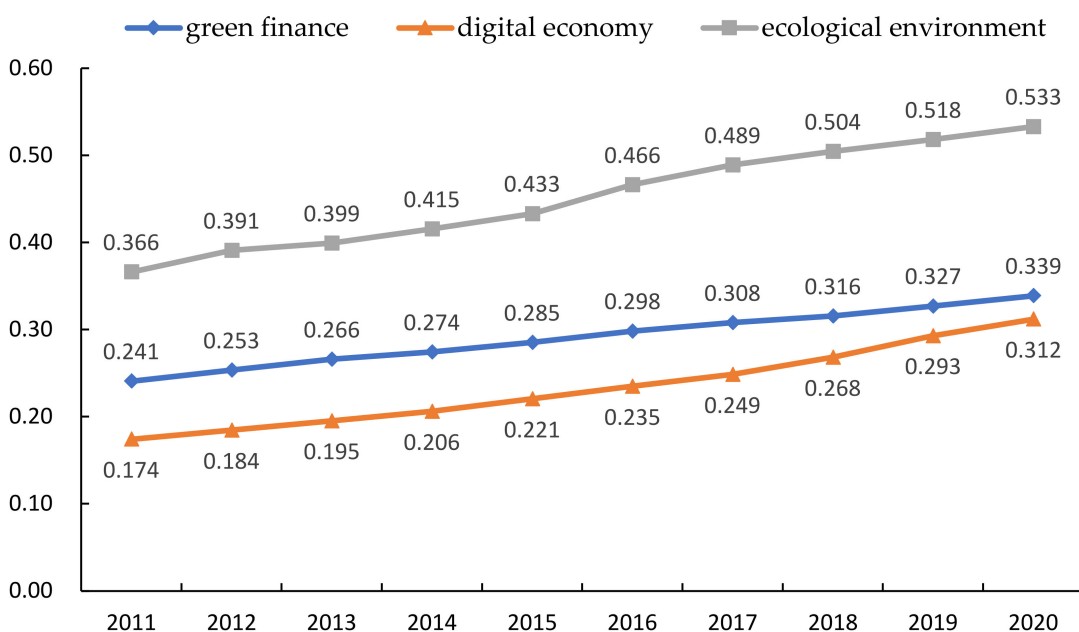

**Figure 1.** The development of green finance, digital economy, and ecosystem environment (source: Stata16, own research results).

*4.2. Analysis of the Spatial and Temporal Evolution of the GDE System*

The coupling coordination degree model in the previous section can be used to calculate the regional GDE system coupling coordination degree (see Table 3 and Figure 2), and the corresponding levels are assigned according to their specific values.

Analyzed from the perspective of time evolution: (1) All regional GDE system coupling correlation degrees reach the grinding stage in 2020, which indicates that green finance, digital economy, and ecological environment are developing synergistically. With the change of time, the coupling correlation degree is a smooth upward trend, but in 2016, the coupling correlation degree decreases slightly in the whole country except for the northeast region. The main reason may be the omission of data related to green bonds. China started to issue green bonds in 2015, which has had a significant promoting effect on the development of green finance, and the total issue size reached USD 30 billion in 2018, occupying 18% of the global green debt volume market issue. However, this indicator is not used in this paper because the data are from 2011, which leads to the underestimation of the GDE system relevance after 2015. (2) Each region's coupling coordination degree exhibits an upward trend. The coupling coordination degree of the entire nation is 0.389 in 2011, representing a mild dissonance, and 0.533 in 2020, which enters the coordination state with an average annual growth rate of 3.56%. The eastern region is 0.478 in 2011, which is on the verge of dissonance, 0.513 in 2013, which enters into coordination for the first time, and 0.640 in 2020, which is primary coordination, with an average annual growth rate of 3.30 percent. The northeastern region is 0.338 in 2011, which is mild disorder, and 0.473 in 2020, which is on the brink of dissonance, with the fastest average annual growth rate of 3.84%. The western region is 0.337 in 2011, which is a mild dissonance, and 0.471 in 2020, which is on the verge of dissonance, with an average annual growth rate of 3.79%. The central region is 0.353 in 2011, which is a mild dissonance, and 0.485 in 2020, which is on the verge of dissonance, with an average annual growth rate of 3.59%.

**Table 3.** The regional coupling coordination degree of the GDE systems.

| Evaluation Index | Region | 2011 | 2012 | 2013 | 2014 | 2015 | 2016 | 2017 | 2018 | 2019 | 2020 |
|---|---|---|---|---|---|---|---|---|---|---|---|
| *C* | Northeast China | 0.496 | 0.529 | 0.533 | 0.529 | 0.522 | 0.538 | 0.548 | 0.545 | 0.557 | 0.565 |
| | Eastern China | 0.613 | 0.624 | 0.657 | 0.666 | 0.687 | 0.685 | 0.692 | 0.706 | 0.731 | 0.758 |
| | Western China | 0.518 | 0.525 | 0.534 | 0.547 | 0.556 | 0.546 | 0.549 | 0.574 | 0.603 | 0.637 |
| | Central China | 0.538 | 0.548 | 0.552 | 0.557 | 0.576 | 0.572 | 0.576 | 0.598 | 0.621 | 0.645 |
| | China | 0.554 | 0.563 | 0.579 | 0.587 | 0.600 | 0.597 | 0.602 | 0.620 | 0.645 | 0.672 |
| *D* | Northeast China | 0.338 | 0.375 | 0.388 | 0.396 | 0.405 | 0.416 | 0.424 | 0.437 | 0.454 | 0.473 |
| | Eastern China | 0.478 | 0.495 | 0.513 | 0.526 | 0.547 | 0.563 | 0.579 | 0.595 | 0.618 | 0.640 |
| | Western China | 0.337 | 0.352 | 0.366 | 0.380 | 0.390 | 0.404 | 0.413 | 0.432 | 0.453 | 0.471 |
| | Central China | 0.353 | 0.366 | 0.378 | 0.387 | 0.403 | 0.415 | 0.430 | 0.445 | 0.464 | 0.485 |
| | China | 0.389 | 0.405 | 0.420 | 0.432 | 0.447 | 0.46C | 0.473 | 0.489 | 0.510 | 0.533 |

Source: Stata16, own research results.

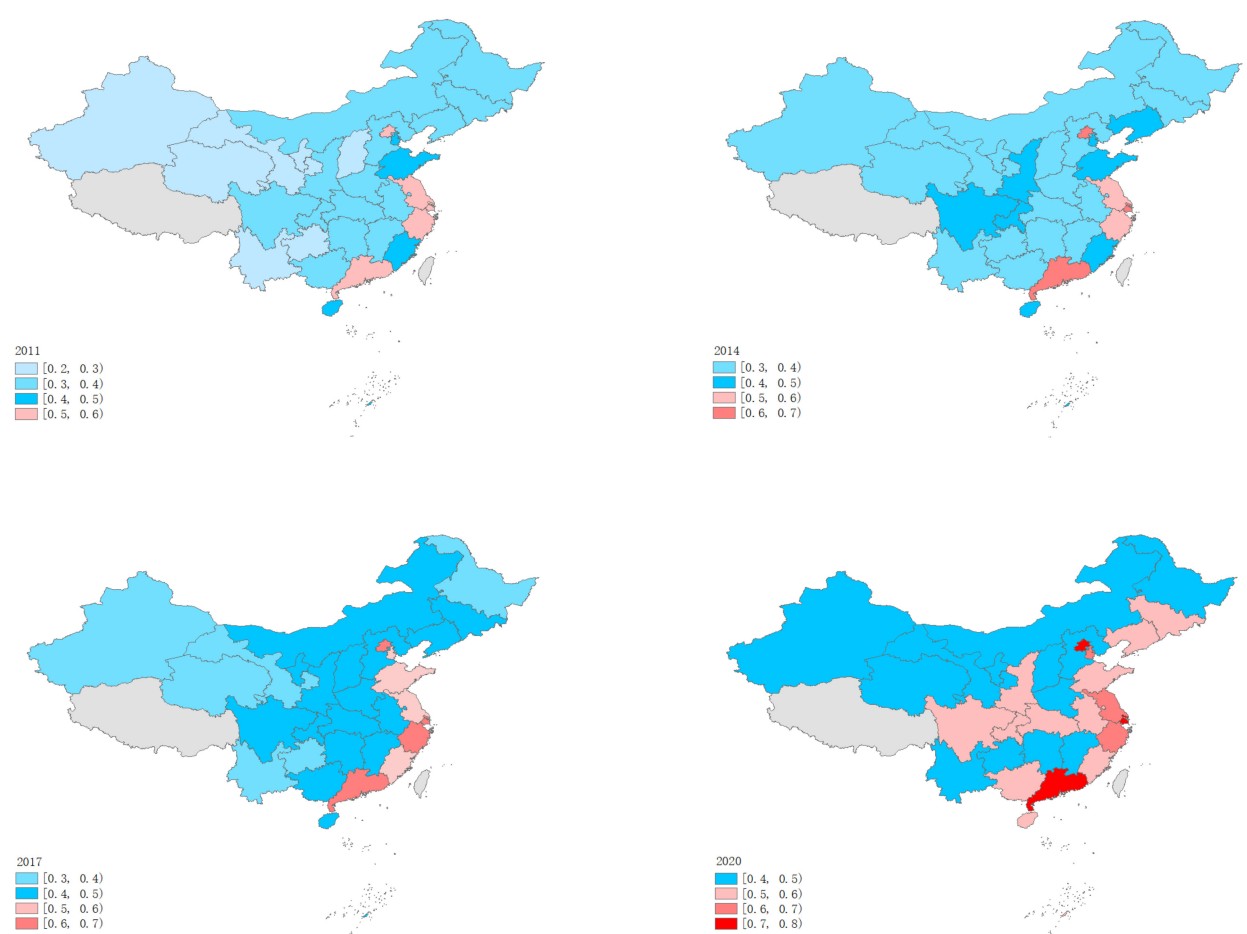

**Figure 2.** The provincial coupling coordination degree of the GDE systems. (Source: Arcmap, own research results).

From the perspective of spatial evolution: (1) There are some differences in the coupling correlation degree of the GDE system in each region, and the coupling correlation degree of the eastern region reaches 0.758 in 2020, which is significantly higher than that of other regions, while the coupling correlation degree of the western and central regions is closer, and that of the northeastern region is lower. (2) The degree of coupling coordination of the GDE system in different regions also varies, with the eastern region's coupling coordination degree being significantly higher than that of other regions, having entered the state of barely coordinated since 2013 and entering the state of primary coordination

in 2019. The coupling coordination degree of the GDE system in the northeast, west, and central regions is comparable at the level of moderate dissonance from 2011 to 2014 and at the level of near dissonance from 2015 to 2020. From a global perspective, the results of the coupling coordination and its coordination degree of the GDE system in each region are essentially the same, and the difference between this result and the level of green finance development in each region is relatively small. Therefore, it can be concluded that although green finance, digital economy, and ecological environment development all have positive promotion effects, the coupling correlation degree of the three is primarily low. For instance, the integrated level of the digital economy and ecological environment is only marginally lower than that of the eastern region, but because of the relatively low level of green finance development, the coupling coordination of the GDE system in this region is not high, indicating that the development of green finance must keep pace with the development of digital economy and ecological environment. The development of the digital economy and ecological environment alone cannot economize. The eastern region has been in the lead, while the northeastern, western, and central regions are gradually catching up with policy guidance and support.

*4.3. Analysis of the Spatial Autocorrelation of the GDE System*

To study the spatial distribution characteristics of GDE system coupling coordination in 30 provinces in China, a spatial autocorrelation model is utilized to study the coupling coordination of the GDE systems.

(1) Global spatial autocorrelation. To quantify the global Moran index of the coupling coordination of the GDE systems, a neighboring spatial weight matrix was constructed. As shown in Table 4, the estimated values of the global Moran index of GDE system coupling coordination for 2011–2020 are all greater than 0 and pass the 5% significance level test, which indicates that the spatial distribution of GDE system coupling coordination level in each province has a significant positive correlation. The global Moran index declined from 0.276 in 2011 to 0.166 in 2016. This is primarily due to the fact that the development of a digital economy can eliminate space-time distance, even regions with a low level of digital economy development can develop rapidly through the rapid sharing and transaction of information, technology, and mode, and there may be a synergistic development trend between provinces, thereby diminishing the phenomenon of regional agglomeration. The Moran index rose from 0.166 in 2016 to 0.199 in 2020, primarily as a result of the establishment of green financial reform and innovation pilot zones in Zhejiang, Guangdong, and other provinces in 2016, as well as the policy dividend resulting from the enhanced coordination of economic and ecological benefits brought about by the development of green finance in each region.

**Table 4.** Global Moran index of coupling coordination of GDE systems (2011–2020).

| Index | 2011 | 2012 | 2013 | 2014 | 2015 | 2016 | 2017 | 2018 | 2019 | 2020 |
|---|---|---|---|---|---|---|---|---|---|---|
| Global Moran index | 0.276 | 0.243 | 0.223 | 0.195 | 0.194 | 0.166 | 0.180 | 0.182 | 0.194 | 0.199 |
| *z* value | 2.688 | 2.549 | 2.473 | 0.526 | 0.547 | 0.563 | 0.579 | 0.595 | 0.618 | 0.640 |
| *p* value | 0.004 | 0.005 | 0.007 | 0.009 | 0.009 | 0.014 | 0.012 | 0.011 | 0.009 | 0.008 |

Source: Stata16, own research results.

(2) Local spatial autocorrelation. Taking into account that the global Moran index reflects the overall correlation degree, this paper selects the average values of the GDE system coupling coordination degree in 2011 and 2020 as the research object to measure the clustering and evolution trend of the GDE system coupling coordination degree in each province to further analyze the local spatial correlation. The calculation can divide each province's spatial clustering types of coupling degree into four quadrants (see Table 5). Most regions fall into the high-high and low-low regions, indicating that the GDE system coupling coordination degree has a clustering effect in both the higher and lower regions and a band-like spatial distribution.

In the HH concentration region: All the eastern provinces are located in this area, mostly in Beijing and Shanghai, and the spatial agglomeration trend is gradually moving towards the eastern coastal region from a scattered state. The main factor for this situation is that the green industry is more developed in Beijing and Shanghai, and thus the development of green finance and digital economy is higher, which leads to the development of environmental protection, which is reflected in the high degree of coupling and coordination in these areas, and the phenomenon of agglomeration. In the LL concentration region: Most of the central and western provinces fall into this area, and the spatial agglomeration trend is gradually increasing and showing a gradual agglomeration trend to the central and western regions. The main factor is that these regions are vast and sparsely populated, and the level of development of green finance and digital economy is relatively low, but the ecological environment is better, thus resulting in a relatively low degree of coupling and coordination, which shows an aggregation effect. The number of provinces falling into the HL and LH regions is relatively small and the spatial distribution is more scattered, mainly because there are large differences and obvious gaps in the coupling coordination degree between this region and the surrounding regions. For example, the coupling coordination degree of the GDE system in Guangdong Province is relatively high, however, the coupling coordination degree of its surrounding provinces is low and fails to spread to Fujian, Guangxi, and the surrounding areas, so Guangdong falls into the HL agglomeration area. In general, the GDE system coupling coordination shows significant distribution characteristics of strong and weak clustering in space. The distribution and quantity of the four quadrant areas remain relatively stable over time, with strong clustering areas primarily located in the eastern region and weak clustering areas primarily distributed in the central and western regions.

**Table 5.** Spatial correlation changes of coupling coordination of GDE systems.

| Space Mode | 2011 | 2020 |
|:---:|:---:|:---:|
| HH | Tianjin, Beijing, Shanghai, Jiangsu, Zhejiang, Fujian, Shandong | Tianjin, Beijing, Shanghai, Jiangsu, Zhejiang, Fujian, Shandong |
| HL | Guangdong, Liaoning | Guangdong, Shaanxi |
| LL | Shanxi, Inner Mongolia, Jilin, Heilongjiang Henan, Hubei, Hunan, Guangxi Chongqing, Sichuan, Guizhou, Yunnan. Shaanxi, Gansu, Qinghai, Ningxia, Xinjiang | Shanxi, Inner Mongolia, Jilin, Heilongjiang Henan, Hubei, Hunan, Liaoning Chongqing, Sichuan, Guizhou, Yunnan. Gansu, Qinghai, Ningxia, Xinjiang |
| LH | Hainan, Hebei, Anhui, Jiangxi | Hainan, Hebei, Anhui, Jiangxi, Guangxi |

Source: Stata16, own research results.

## 5. Analysis of Driving Factors of the Coupling Coordination of the GDE System

The coupling coordination of green finance, digital economy, and ecological environment in each region is a relatively complex mechanism, which is the result of the joint action of multiple factors. To further explore its driving factors, a grey correlation model of panel data is selected to analyze and study the driving factors of the coupling coordination development of the GDE system. Considering the development of the GDE system comprehensively and referring to the research results of Fu [16], the following three factors are selected as the driving factors: (1) Industrial structure: Regional distinctions in the industrial structure are a direct result of the disparate levels of industrial structure development. The relationship between the industrial structure and the coupling and coordination of the GDE system is close. The industrial structure can improve regional economic development, promote the development of industrial digitalization and digital industrialization, then promote the development of digital economy, as well as promote the construction of ecological civilization, thereby enhancing the ecological environment and promoting the growth of the green industry. The ratio of tertiary industry output to secondary industry output is utilized to measure specific indicators. (2) Government regulation: Government financial support, policy guidance, and institutional regulations play a crucial regulatory

role in the coupling coordination development of the GDE system and are the coupling coordination of the GDE system's guarantee force, as measured by per capita local financial expenditures in this paper. (3) Economic development: In this paper, GDP per capita is used as a specific indicator of the relationship between the level of economic development and the level of development of green finance and the digital economy, which not only improves people's quality of life and enhances the development of digital technology, but also provides financial support to optimize the ecological environment. The specific steps are as follows:

Step 1. The coupling and coordination degree of green finance, digital economy, and ecological environment of each province, region, and city as a reference sequence $y(k)$. The industrial structure, government regulation, and economic development are used as the comparative sequence $x_i(k)$, where $x_i(k)$ denotes the driving factors, and $k$ is the specific indicator data.

Step 2. Initialization of data for each indicator:

$$x_i(k) = \frac{x_i(k)}{x_i(1)} \tag{14}$$

Step 3. Calculate the number of correlation coefficients:

$$r_i(k) = \frac{\min\limits_{i} \min\limits_{k} |y(k) - x_i(k)| + \rho \max\limits_{i} \max\limits_{k} |y(k) - x_i(k)|}{|y(k) - x_i(k)| + \rho \max\limits_{i} \max\limits_{k} |y(k) - x_i(k)|} \tag{15}$$

where the resolution factor $\rho$ takes the value of 0.5.

Step 4. Calculate the gray correlation:

$$r_i = \frac{1}{m} \sum_{k=1}^{m} r_i(k) \tag{16}$$

Step 5. Correlation degree classification level. $0 < r_i \leq 0.4$ is a weak correlation, $0.4 < r_i \leq 0.6$ is a medium correlation, $0.6 < r_i \leq 0.8$ is a strong correlation, and $0.8 < r_i \leq 1.0$ is a strong correlation.

According to Table 6, the correlation between the three factors and the level of GDE system coupling and coordination is above 0.6, which is a strong correlation, indicating that the drivers are closely related to the development of GDE system coupling and coordination, and there are differences in different regions.

The average correlation between industrial structure and coupling and coordination of the GDE system is 0.6583. The transformation and upgrading of industrial structures can expand the scale and development level of the regional economy and improve the development level of the digital economy. Concurrently, with the continuous optimization of industrial structure, the energy structure, which was formerly dominated by fossil energy, can be significantly improved, thereby reducing regional carbon emissions and promoting the improvement of green finance and the ecological environment. The most significant impact is 0.6854 for the eastern region, which is inextricably linked to the region's robust industrial structure. The northeast region follows while the central and western regions are closer.

The average correlation between government regulation and coupling and coordination of the GDE system is 0.6210, indicating that the implementation of active government financial support, policy guidance, and institutional improvement can promote the coupling and coordination of the GDE system effectively. Central and western regions are above the mean, while northeastern and eastern regions are below it. This indicates that the influence of government regulation on the coupled and coordinated development of central and western regions is significantly greater than that of other regions, primarily

because the socioeconomic development level of central and western regions is relatively low due to their location conditions and historical development background, and thus the influence of policy regulation on the coupled and coordinated development of the GDE system is more effective.

The highest mean value of correlation between economic development and GDE system coupling coordination is 0.7362, indicating that economic development is the key factor of GDE system coupling coordination development, among which the eastern region is significantly higher than the mean value, which is mainly because the eastern region occupies the dominant position in the national economic development, and is better than other regions in both infrastructure construction and employment and income, and thus the economic agglomeration effect formed promotes the coupling coordination development of the GDE system. The western region has the lowest influence of 0.6926 due to its relatively weak economic development and should increase the resource inclination and policy support to the central and western regions, to realize the comprehensive and coordinated development of each region.

**Table 6.** Correlations of the GDE system coupling coordination drivers.

| Region | Industry Structure | Government Regulation | Economic Development |
|---|---|---|---|
| China | 0.6583 | 0.6210 | 0.7362 |
| Northeast China | 0.6523 | 0.5864 | 0.7485 |
| Eastern China | 0.6854 | 0.5380 | 0.7853 |
| Central China | 0.6274 | 0.7301 | 0.7459 |
| Western China | 0.6301 | 0.7117 | 0.6926 |

Source: Stata16, own research results.

## 6. Conclusions and Policy Suggestions

This paper investigates the spatio-temporal evolution characteristics of coupling coordination between GDE systems and driving factors. The primary conclusions of this paper consist of four components. Firstly, the comprehensive levels of green finance, digital economy, and ecological environment in China all maintain stable growth, with the digital economy developing at the quickest rate among the three. Secondly, the coupling coordination degree of the three regions exhibits a gradual upward trend, and the entire nation enters the coordination state in 2019, with the eastern region taking the lead in the primary coordination state and the northeast, western, and central regions on the verge of dissonance. Thirdly, there are significant global and local autocorrelations in the coupling coordination degree of GDE systems with positive global spatial autocorrelation and local spatial autocorrelation, and strong and weak spatial clustering characteristics. Finally, the coupling coordination of the GDE system is the result of multiple factors, and the primary motivating factors vary between regions. Economic growth and industrial structure have the most impact on the eastern region, while government regulation has the most impact on the central region.

Based on the preceding research methodology and findings, we propose the following policy suggestions:

Firstly, improve the green financial system. Government should invest more in the development of the renewable financial system, optimize and improve the green financial system, strengthen the policy service mechanism and the construction of the legal environment, optimize the construction of the legal and regulatory environment of China's green finance, promote the governance environment in accordance with the law, and improve the relevant legal measures in accordance with the law. Concurrently, the government should establish a solid system for green financial innovation and a market system, optimize the industrial structure, and expedite the development of green industries in the primary, secondary, and tertiary sectors.

Secondly, coordinate the synergy of green finance and digital economy to enhance the benefits of environmental governance. To achieve the integration of domestic enterprise finance and environmental governance and to realize the goal of symbiotic development,

actively steer the transformation of domestic enterprises toward environmental protection, resource conservation, sustainable development, and eco-friendly green development. Government should simultaneously lead the flow of green finance to the level of ecological environmental protection, ecological construction, green industry, and environmental governance, enhance science and technology, strengthen the influence of advanced science and technology level, and expand the reach and popularity of science and technology for environmental protection.

Thirdly, based on the coupling coordination of its green finance, digital economy, and ecological environment, each region should devise a development approach that takes into account its unique characteristics. With its robust economic strength, the eastern region adheres to the new concept of development, rapidly develops green industries, realizes the transformation of industrial digitalization and digital industrialization, and focuses on green and low-carbon technology innovation. For the northeast, central, and western regions, appropriate policies and financial support should be provided to achieve comprehensive environmental improvement, reduce resource consumption, and increase resource use efficiency by transforming the mode of economic development, increasing wastewater and waste gas treatment, and increasing investment in environmental pollution treatment. They should coordinate the relationship between GDE systems to promote the growth of green finance and the digital economy while enhancing the ecological environment continuously.

This paper contributes to the literature on the coupling coordination of green finance, digital economy, and ecological environment, but there are still some limitations. Firstly, due to the difficulty of data acquisition, the GDE systems in Hong Kong, Macao, Tibet, and Taiwan are not analyzed in this paper. Secondly, this paper utilizes provincial-level data, which can be refined to include cities and counties, as well as spatial comparisons of the system's development trends. Thirdly, this paper selects industrial structure, government regulation, and economic development as driving factors of the coupling coordination of GDE systems. It is strongly suggested to investigate new driving factors and determine how they influence the coupling coordination of GDE systems. Additionally, regardless of the traditional or modified coupling coordination model, there are still problems of subjectivity in index construction, volatility of coupling results, and reliability problems of incomparability. Therefore, how to improve the reliability and stability of the model is a question worthy of further discussion.

**Author Contributions:** Conceptualization, L.Z. and Y.Z.; methodology, L.Z. and Y.Z.; data collection and analysis, L.Z.; writing—original draft, L.Z.; writing—review and editing, Y.Z. All authors have read and agreed to the published version of the manuscript.

**Funding:** This research received no external funding.

**Institutional Review Board Statement:** Not applicable.

**Informed Consent Statement:** Not applicable.

**Data Availability Statement:** The data used in this research is available upon request.

**Conflicts of Interest:** The authors declare no conflict of interest.

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
