# Peer review of "Research on the Coupling Coordination of Green Finance, Digital Economy, and Ecological Environment in China"

_sustainability, doi:10.3390/su15097551_

Round 1
Reviewer 1 Report
This investigation analyzes the coupling coordination of green finance, digital economy, and ecological environment and constructs an evaluation index system of coupling coordination degree. This is an innovative research topic and has important significance for the development of regional digital finance. However, before being received, it is necessary to clarify the following:
1:In line 165, is there a formula for Wj and the physical meaning of the representation?
2:Both Equation 4 and Equation 5 show the calculation expression of C. Which formula is more representative for this investigation?
3:Can the author explain the trend of the curve falling first and then rising in Figure 2?
4:It is recommended to supplement the following references (①10.1016/j.molliq.2023.121394 ②10.1007/s40948-022-00396-0 ③10.1007/s11356-023-26279-9) on the relationship between the digital economy and the ecological environment.
Author Response
Dear reviewers,
Thank you for your precious comments and advice. Those comments are all valuable and very helpful for revising and improving our paper, as well as the important guiding significance to our researches. We have studied comments carefully and have made corrections. Please see the attachment.
We would love to thank you for allowing us to resubmit a revised copy of the manuscript and we highly appreciate your time and consideration.
Sincerely.
Lifang Zhang and Yuexu Zhao.

Reviewer 2 Report
This paper requires extensive revision before it can be published:
1. The abstract is a summary of the whole paper. In the abstract, in addition to a summary of the background, research methods and findings of this paper, it also needs to contain the suggestions to the research problems. It is recommended that the author add this content at the end of the abstract.
2. In the general writing order, the second part of the paper should be a literature review. I hope the author can add this part to make the structure of the paper more complete.
3. In the introductory part of this paper, especially when introducing the research background, it is recommended to first elaborate on the international research background, and then on the Chinese research background, so that the logic of the introductory part can be clearer and more complete.
4. In this paper, the author should clarify the innovative points or aspects of the article that are different from the research directions (or research methods) studied by the previous authors, and explain in which aspects they have been improved.
5. At the end of the introduction, it is suggested that a brief introduction to the content of each section below be added.
6. The grammar of the paper needs to be further revised to make it more fluent and complete. In addition, the tables and pictures in the paper need to be more standardised, especially the pictures should not be too blurred.
Author Response

(The authors gave the same response as above.)

Reviewer 3 Report
The article fits within the scientific framework of the journal and can reach a wide audience. Nevertheless, it is not without shortcomings. It is recommended: (1) describing the structure of the article in the introduction; (2) eliminating editorial errors (the wording "error" appears in the text instead of references to figures or tables); (3) placing sources under tables and figures; (4) citing other articles that have been published in the journal and that correspond to the issue taken up; (5) indicating in summary the limitations of the research carried out, including the methods used, and proposing future research directions; (6) distinguishing a section with a literature review and citing the results of other authors in the field.
Author Response

(The authors gave the same response as above.)

Reviewer 4 Report
Dear Authors
The article should be corrected/supplemented:
· General literature review, key items of world literature on this topic is missing. The authors mainly use their native literature.
· Theoretical and practical conclusions and implications are presented in the theoretical language appropriate to the analysis. This fact makes it difficult to understand and assess the usefulness of such a study. It is worth highlighting the practical aspect of the article more.
· The article should highlight the limitations of the research carried out and the conclusions drawn.
· Technical notes, do not use underlining in the article (page 1).
Author Response

(The authors gave the same response as above.)

Reviewer 5 Report
The article "Research on the Coupling Coordination of Green Finance, Digital Economy, and Ecological Environment in China" is current and of interest to the scientific environment both through the pragmatic approach and the logic of argumentation. The research has a high level of replicability. Several aspects require adjustments to give the article a publishable character:
1) Authors must carry out a serious review of bibliographic sources and citations in the text (see lines 123, 189, 209, 231, 292, 304, 376).
2) Tables and figures must be cited in the text.
2) Table 3 must be interpreted and aranged. The interpretation must present synthetically the evolution of the GDE system. It would be interesting the presenting dynamic maps by correlating the information with those in table 2.
3) Figure 2 needs to be improved. It would be useful to structure the abscissa by levels of representativeness of the index. The figure is neither interpreted nor cited in the text (see point 2).
4) The conclusions must briefly present the achievement of the objectives of the study, its replicable nature and the main limits and research directions.
5) It would be good to avoid expressions like "our government", "our ecological environment" which reduce the international character of the research.
Author Response

(The authors gave the same response as above.)

Round 2
Reviewer 2 Report
I am satisfied with the revised version.
The data in this manuscript are reliable, the research methods used are reasonable, and the results obtained are robust and interesting.
I think this paper is suitable for publication in Sustainability now.
Author Response
We appreciate the reviewer’s positive evaluation of our work.
Reviewer 3 Report
Authors should indicate sources under tables and figures (e.g. own elaboration or own elaboration based on...) before publishing the article, and indicate in the conclusion the limitations of the research methods used.
Author Response
Dear reviewers,
Thank you again for your precious comments and advice. Those comments are all valuable and very helpful for revising and improving our paper, as well as the important guiding significance to our researches. We have studied comments carefully and have made corrections. Please see the attachment.
We would love to thank you for allowing us to resubmit a revised copy of the manuscript and we highly appreciate your time and consideration.
Sincerely.
Lifang Zhang and Yuexu Zhao.

Reviewer 5 Report
The manuscript has been sufficiently improved. Congratulations.
Author Response

(The authors gave the same response as above.)
